# Homologous Recombination Deficiency in Ovarian Cancer: from the Biological Rationale to Current Diagnostic Approaches

**DOI:** 10.3390/jpm13020284

**Published:** 2023-02-02

**Authors:** Alessandro Mangogna, Giada Munari, Francesco Pepe, Edoardo Maffii, Pierluigi Giampaolino, Giuseppe Ricci, Matteo Fassan, Umberto Malapelle, Stefania Biffi

**Affiliations:** 1Obstetrics and Gynecology, Institute for Maternal and Child Health, IRCCS Burlo Garofolo, 34137 Trieste, Italy; 2Veneto Institute of Oncology, IOV-IRCCS, 35128 Padua, Italy; 3Department of Public Health, University of Naples Federico II, 80131 Naples, Italy; 4Department of Medicine (DIMED), University of Padua, 35128 Padua, Italy; 5Department of Medical, Surgical and Health Science, University of Trieste, 34149 Trieste, Italy

**Keywords:** ovarian cancer, homologous recombination deficiency (HRD), homologous recombination repair (HRR), next-generation sequencing (NGS), poly (ADP-ribose) polymerase inhibitor (PARPi), test, assay

## Abstract

The inability to efficiently repair DNA double-strand breaks using the homologous recombination repair pathway is defined as homologous recombination deficiency (HRD). This molecular phenotype represents a positive predictive biomarker for the clinical use of poly (adenosine diphosphate [ADP]-ribose) polymerase inhibitors and platinum-based chemotherapy in ovarian cancers. However, HRD is a complex genomic signature, and different methods of analysis have been developed to introduce HRD testing in the clinical setting. This review describes the technical aspects and challenges related to HRD testing in ovarian cancer and outlines the potential pitfalls and challenges that can be encountered in HRD diagnostics.

## 1. Homologous Recombination Deficiency (HRD)

As scientists continue to investigate and comprehend the foundations of cancer genomics, they uncover wider molecular fingerprints present in various types of cancer. Homologous recombination deficiency (HRD) is one of these signatures that appears to be gaining significance in the biology of multiple cancers including ovarian, breast, pancreatic, uterine, genitourinary, colorectal, gastrointestinal, hepatocellular carcinoma, biliary tract cancer, sarcoma, melanoma and prostate malignancies [1,2].

HRD is a complex genomic signature that emerges when a cell is unable to repair broken double-stranded DNA via the homologous recombination repair (HRR) pathway [1] (Figure 1). Cells must be able to repair DNA damage to sustain genomic stability and cell function. This capacity maintains chromosome integrity and keeps cells alive.

Multiple genes mediate the HRR pathway, with *BRCA1* and *BRCA2* playing crucial roles (Table 1) [2,3,4,5,6]. If the HRR pathway is compromised, double-strand breaks are not repaired or fixed through the error-prone non-homologous end-joining (NHEJ) pathway. These conditions may cause genomic instability in the form of genomic scarring, resulting in malignant transformation [7].

HRD-related genomic markers are also known as “scars”. Genomic scars can be defined as aberrations that cause structural changes in chromosomes. The most significant genomic scars are loss of heterozygosity (LOH) [8], telomeric-allelic imbalance (TAI) [9], and large-scale state transitions (LSTs) [10] (Table 2). When these three abnormalities of the genome are looked at together, they give a genomic instability score (GIS) that can be used to tell the status of HRD.

HRD status can be assessed by evaluating the presence of causal genes (*BRCA* and other HRR genes) and/or the genomic scar. Several tests for HRD valuation status are nowadays available, each with its specific criteria [11]. Some assays identify genomic instability based strictly on the percentage of LOH. Accumulating evidence suggests that evaluation of all three genomic aberrations (LOH, TAI, LST) may improve the identification of HRD-positive samples [12,13]. The result offers sensitive and reliable evaluation of HRD status and other cancer-associated genomic variants that may be present in a sample.

## 2. HRD in Ovarian Cancer

HRD is an emerging biomarker with both predictive and prognostic value in high-grade serous ovarian carcinoma (HGSOC). The Cancer Genome Atlas (TCGA) project has revealed that about 50% of HGSOC show HRD, which has a variety of causal factors [14], many of which are still not well understood. Typically, loss of function mutations and epigenetic modification in *BRCA1/2* or genes encoding other fundamental players in the HRR pathway (Table 1), such as *ATM*, *BARD1*, *BRIP1*, *H2AX*, *MRE11*, *PALB2*, *RAD51*, *RAD51C/D*, *RPA*, and Fanconi Anemia Complementation Group genes, have been identified as important causative factors of HRD in HGSOC [15].

Poly (ADP-ribose) polymerase inhibitors (PARPis) were developed based on their projected synthetic lethality in HRD-positive tumor cells. PARP1 (poly(ADP-ribose) polymerase 1) is a multifunctional enzyme [16] well documented for its function in the recovery of single-strand DNA breaks via the base excision pathway [17]. PARPis bind to PARP1 at single-strand DNA breaks, avoiding effective repair and causing DNA-protein crosslinks to be processed into double-strand breaks (DSBs), which results in increased genomic instability and cellular death in BRCA1/2-mutant or other HRD-affected cells that are already defective in their DSB repair capacity. HRD has thus been identified as a possible prognostic biomarker for PARPi therapy in HGSOC, breast, pancreatic, and prostate malignancies based on these findings [18,19,20,21].

The *BRCA* test can be performed on peripheral blood and neoplastic tissue, highlighting genetic and somatic variants. All patients with non-mucinous or non-borderline ovarian cancer should be tested for the somatic *BRCA* mutation status at the time of disease diagnosis [22]. In the presence of a positive test on the tumor, a genetic test must always be performed on a blood sample to distinguish germline mutations, which require counselling and genetic testing in family members, from somatic ones [23,24].

HRD has been observed not only in ovarian cancer patients with germline or somatic *BRCA1* or *BRCA2* mutations but also in those with epigenetic silencing of *BRCA1* or loss of function of other genes, such as *ATM*, *ATR*, *BARD*, *BRIP*, *EMSY*, *PALB2*, *RAD51*, and Fanconi Anemia Complementation Group genes [25,26,27,28,29,30]. 

These patients have a “BRCAness” phenotype, similar to that of patients with germline *BRCA1* or *BRCA2* mutations, which includes serous histology, high response rates to first and subsequent lines of platinum-based therapy, long duration between recurrences, and best overall survival (OS) [31,32,33,34]. The presence of the BRCAness profile identifies a subgroup of patients with sporadic ovarian cancer with a better prognosis and excellent responsiveness to platinum and PARPi agents [35]. PARPis are approved by the European Medicines Agency (EMA) and the US Food and Drug Administration (FDA) for the management of HGSCs in different clinical settings: (i) as first-line maintenance therapy to treat women who had a complete or partial response to their most recent treatment with chemotherapy (platinum chemotherapy); (ii) second-line maintenance therapy for platinum-sensitive relapsed disease (regardless of BRCA mutation or other HRD test status); and (iii) monotherapy treatment for HGSC with a BRCA mutation (olaparib/rucaparib) or a positive HRD test (niraparib) after two lines of therapy [36]. HRD testing is performed to determine if a patient is eligible for treatment. NCCN Guidelines say that all patients with ovarian cancer that has been confirmed by a pathologist should get HRD testing [37]. It is recommended by the ESMO guidelines [36] that the following tests be performed in the first-line maintenance setting: (i) germline and somatic BRCA mutation testing to identify HGSC patients who should receive a PARPi; (ii) an HRD test to establish the magnitude of benefit conferred by PARPi use in BRCA wild-type HGSC; and (iii) an HRD test to identify the subgroup of BRCA wild-type patients who are least likely to benefit from PARPi therapy. It is feasible to utilize BRCA mutation testing and HRD testing in the platinum-sensitive relapse maintenance context to estimate the expected extent of the PARPi benefit for the purpose of considering the risks and benefits of maintenance treatment. Three PARP inhibitors have received FDA approval for ovarian cancer maintenance therapy: olaparib, niraparib, and rucaparib [38,39,40]. Several trials examined the efficacy of PARPis in first-line ovarian cancer maintenance therapy (Table 3). 

The SOLO-1 trial findings in 2018 led to olaparib being approved as a first-line maintenance treatment in patients with BRCA1/2 mutations by the EMA and the FDA, creating a new standard of care. Then, in 2019, the results of three phase III trials (PRIMA, PAOLA-1, and VELIA) that looked at the use of first-line PARP inhibitors beyond patients with BRCA1/2 mutations and as combination strategies were presented. This resulted in the recent approval of niraparib maintenance independent of biomarker status as well as olaparib in conjunction with bevacizumab in HRD-positive advanced ovarian cancer [41]. Following response to platinum-based chemotherapy, there is a substantial and clinically relevant advantage to adding PARP inhibitor maintenance treatment (alone or in combination with bevacizumab) in patients with HRD-positive tumors. Due to the low efficacy of PARP inhibitors in HR-proficient individuals and their worse prognosis, there is an urgent need for alternative therapeutic regimens in this patient subgroup [41].

**Table 3 jpm-13-00284-t003:** Clinical trials evaluating PARPis in first-line ovarian cancer maintenance therapy.

Trial	Drug	Efficacy Data	Ref.
ARIEL2	Rucaparib	Response rates: 69% for BRCA-mutated tumors, 39% for BRCA wild-type and high LOH tumors, and 11% for BRCA wild-type and low LOH tumors (*p* < 0.0001).	[42,43]
ARIEL3	Rucaparib	Improved PFS compared with placebo in both patients with mutated BRCA (16.6 versus 5 months, HR = 0.23, 95% CI = 0.16–0.34) and those with HRD (13.6 versus 5.4 months, HR = 0.32, 95% CI = 0.24–0.42), and in the whole population (10.8 versus 5.4 months, HR = 0.36, 95% CI = 0.30–0.45) including LOH negative patients.	[44]
NOVA	Niraparib	Niraparib was associated with better median PFS in all subgroups, with a more substantial benefit in the gBRCA cohort (21.0 versus 5.5 months, HR = 0.27; 95% CI = 0.17–0.41) than in the non-gBRCA cohort with HRD (12.9 versus 3.8 months, HR = 0.38; 95% CI = 0.24–0.59) and the entire non-gBRCA cohort (9.3 versus 3.9 months, HR = 0.45; 95% CI = 0.34).	[45]
Study19	Olaparib	Improvement in OS in patients with BRCA-mutated recurrent ovarian cancer who received olaparib maintenance therapy following platinum-based chemotherapy (29.8 versus 27.8 months, HR = 0.73; 95% CI = 0.55–0.95). This occurrence represents a secondary endpoint of the trial. The results support previously reported benefits of olaparib in PFS compared to placebo, which is the primary trial endpoint (8.4 versus 4.8 months, HR = 0.35; 95% CI = 0.25–0.49).	[46,47]
SOLO-1	Olaparib	The SOLO-1 trial demonstrated a long-term PFS advantage for olaparib versus placebo in the first-line maintenance treatment of patients with HGSOC with a germline or somatic mutation in BRCA1/2 who had a complete or partial response after platinum-based chemotherapy. Data revealed olaparib reduced the risk of disease progression or death by 67% (based on an HR of 0.33; 95% CI = 0.25–0.43) and increased PFS to a median of 56.0 months compared with 13.8 months of placebo.	[48,49]
SOLO-2	Olaparib	Results from the phase III SOLO-2 trial demonstrate an improvement in PFS in patients with platinum-sensitive relapsed ovarian cancer and gBRCA mutations treated with olaparib compared to placebo in the maintenance setting: PFS (19.1 versus 5.5 months, HR = 0.30; 95% CI = 0.22–0.41). Although there was no statistically significant difference in OS, the results supported the use of olaparib for maintenance in these patients.	[50,51,52]
PRIMA	Niraparib	In the HRD population, maintenance therapy with niraparib led to a reduction in the risk of progression or death by 57% (HR = 0.43; 95% CI 0.31–0.59), whereas in the intention-to-treat population, the risk reduction was 38% (HR = 0.62; 95% CI 0.5–0.76). In the HR-proficient subgroup maintenance therapy with niraparib led to 32% reduction in the risk of progression or death (HR = 0.68; 95% CI, 0.49–0.94).	[20,49]
VELIA	Veliparib	Across all patient subgroups, PFS for the combined induction and maintenance phases in the veliparib arm was 23.5 months versus 17.3 months in the placebo arm (HR = 0.68; 95% CI 0.56–0.83). The benefit was most evident for those with BRCA mutations. In this group, the median PFS was 34.7 months, compared with 22.0 months for veliparib and placebo, respectively (HR 0.44; 95% CI 0.28–0.68), whereas in the HRD cohort, PFS was 31.9 versus. 20.5 months (HR = 0.57; 95% CI: 0.43–0.76). Unfortunately, veliparib is not FDA-approved.	[53]
PAOLA-1	Olaparib	Results showed that olaparib in combination with bevacizumab reduced the risk of disease progression or death by 41% and improved PFS in the intention-to-treat population with a median of 22.1 months compared with 16.6 months in patients treated with bevacizumab monotherapy (HR = 0.59; 95% CI 0.49–0.72). Subgroup analysis also highlighted an essential synergistic effect of the combination in patients with BRCA mutation and HRD, with a mean PFS of 37.2 months and an HR of 0.31 (95% CI 0.20–0.47) and 0.33 (95% CI 0.25–0.45), respectively. However, adding olaparib to bevacizumab showed almost no effect in the HRD-negative or unknown HRD status subgroup, where the median PFS was 16.9 months in the experimental arm versus 16 months in the control arm. These data support the hypothesis of the synergistic effect of olaparib and bevacizumab in BRCA mutant and HRD patients, underscoring the importance of HRD status as a novel prognostic factor and predictor of response to PARP inhibitors.	[49,54]

Abbreviations: PFS: progression-free survival; HR: hazard ratio (HR); CI: confidence interval (CI); gBRCA: germline BRCA mutation.

Ovarian carcinomas in germline BRCA-mutated women have a more elevated mutational burden and a higher number of neoantigens that stimulate the recruitment of tumor-infiltrating lymphocytes (TILs) [55,56,57]. These tumors revealed increased numbers of CD3+ and CD8+ TILs and elevated expression of PD-1 and PD-L1 and may therefore represent a subset of malignancies particularly sensitive to treatment with immune checkpoint inhibitors alone or in combination with PARP inhibitors and/or chemotherapy [58].

## 3. HRD Assays

This clinical tests for HRD aim to predict the presence of HRD based on genomic features [59]. Evaluation of HRD in the clinical setting is an essential tool that has the potential to aid patient selection for PARPi and other DNA-damaging agents in ovarian cancer, but understanding the details of these tests and their limitations is critical to ensuring their optimal clinical application [59]. HRD testing is FDA-approved only for ovarian cancer; however, it seems fundamental to defining an appropriate therapy for prostate, pancreatic, and breast cancers. For this reason, it is necessary to perform this test in-house in these neoplasms. Still, it is a significant challenge to make tests that can find the HRD phenotype of cancers (HRD tests) and predict sensitivity to PARPi so that patients who are most likely to benefit from this therapy can be chosen [60]. 

HRD tests are classified into three types: germline or somatic variants of genes that are part the HRR pathway; genomic scars or mutational profiles showing characteristics of genomic instability; and HRR functional status testing (Figure 2) [61]. 

The effects of a disrupted HRR pathway are studied by examining the genome for signs of genomic aberrations. Numerous investigations, including ovarian and breast cancer have revealed genetic profiles or markers of instability related to an HRD phenotype. These signs of instability may consist of: (i) the total count of telomeric imbalances (TAI), which are parts with allelic disparities that encompass the sub-telomere but do not pass the centromere [9], (ii) genomic profiles of loss of heterozygosity (LOH), which are parts of intermediate size (> 15 MB and < whole chromosome) [8], and (iii) large-scale transitions (LST), which are chromosome breaks (inversions, deletions, or translocations) [11]. These methods assess the status of HRD-related genomic markers (also known as “scars”), considered the result of error-prone DNA repair via different mechanisms [11].

### 3.1. Mutations of Genes in the HRR Pathway

Both the BRCA1 and BRCA2 genes play crucial roles in the HRR pathway. A reduction in the activity of the BRCA genes is one of the processes that has received the most attention among the HRD-causing factors in tumor cells [11].

All newly diagnosed epithelial ovarian cancer patients should have germline and somatic *BRCA* tests. Pathogenic *BRCA1/2* mutations are the most common cause of hereditary ovarian cancer, accounting for 20% of all cases [62]. BRCA1 and BRCA2 work independently to maintain genomic integrity, playing critical roles in the DNA repair mechanism of HR [63]. Patients with ovarian cancer who would benefit from PARPi maintenance therapy can be identified using germline and tumor *BRCA* testing. Because 5% of germline *BRCA*-mutated patients test negative for tumor *BRCA*, a somatic test cannot replace a germline test. In patients with negative germline *BRCA* tests, the somatic *BRCA* tests can reveal an additional 6–7% of patients with *BRCA* mutations that have arisen during cancer development or progression [26]. There is evidence that offering tumor tissue testing first, followed by germline testing for pathogenic or likely pathogenic variants (PV/LPV) detected in the tumor or additional NGS germline testing in cases of positive personal or family history for hereditary breast and/or ovarian cancer, would allow nearly all PV/LPV to be detected as quickly as parallel blood and tumor testing while significantly reducing the cost and workload involved [23]. 

About 30% of HGSOC have HRR pathway alterations, according to the Cancer Genome Atlas (TCGA) [26]. Members of the Fanconi anaemia family, such as *RAD51C*, *RAD51D*, and *BRIP1*, have germline or homozygous somatic mutations that increase susceptibility to HGSOC [64,65], and pre-clinical studies have proven that defects in these genes, as well as probably other HRR-associated genes, such as *ATM*, *CHEK1*, *CHEK2*, and *CDK12*, also impart sensitivity to DNA repair inhibition [34,64,66]. Moreover, the amplification of *EMSY* (a BRCA2-interacting transcriptional repressor) is associated with HRD, whereas *CCNE1* amplification is associated with HR proficiency and a poor prognosis [67]. Clinical trials proved that somatic mutations in non-BRCA HRR genes offer a PFS and overall survival advantage similar to BRCA mutations in patients receiving platinum treatment. However, because non-BRCA HRR mutations are relatively uncommon, these studies combined all HRR genes. In contrast, other data on individual HRR genes is anecdotal, making it difficult to evaluate the relevance of any specific HRR gene at this time [36].

### 3.2. Genomic Scars or Mutational Markers Showing Profiles of Genomic Instability

The prevalence of HRD genomic tests currently in use were created using SNP-based microarray technologies to assess somatic copy number variation (CNV). Three studies published SNP-based CNV assays that predicted BRCA status by measuring large-scale transitions (LST) [10], loss of heterozygosity (LOH) [8], or the number of subchromosomal areas with allelic imbalance extending to the telomere [9]. Subsequent research indicated that integrating the data obtained with these assays improved the capacity to discriminate between HRR-competent and deficient tumors [12]. Two commercially available tests combine tumor BRCA status testing with either an estimate of the percentage of genomic sub-chromosomal LOH (FoundationOne, Foundation Medicine) or a genomic instability score based on the overall score sum of TAI, LST, and LOH (myChoice HRD test, Myriad Genetics) (Figure 2).

GIS is the only genomic scar assay studied in first-line randomized controlled trials [36]. Preclinical research suggests that mutation-based tests that rely on data from various mutation types could outperform current scar assays. Nevertheless, one significant restriction is the need for fresh frozen samples, whereas the prevalence of trial materials is formalin-fixed paraffin-embedded (FFPE). There is currently insufficient data demonstrating the accuracy with which mutational markers derived from whole genome sequencing may be utilized to determine the PARPi response in HGSOC [36].

### 3.3. Functional Assays of HRD

All commercially available HRD tests indicated in Figure 2 are DNA-based, rating mutagenesis events that occurred during tumor evolution. However, selection pressure from systemic treatments may promote resistance mechanisms, particularly in the metastatic context, which is likely to impede accurate HRD identification. The most typical experimental approach for estimating HRR functional status has been to quantify the nuclear RAD51, an associated HR protein (a DNA recombinase) that facilitates DNA strand invasion into the sister chromatid, a step facilitated by the BRCA1/PALB2/BRCA2 complex (Figure 2). In ovarian and breast cancer experimental models, including ex vivo cultures derived from ascites or solid HGSOC, decreased DNA damage-induced nuclear RAD51 foci have been associated with BRCA1 or BRCA2 gene deficiencies and PARPi responses [68]. The RAD51 assay has recently proved feasible in routine FFPE tumor tissues, particularly in selecting patients with breast cancer who may be responsive to PARPi [69]. In patients with ovarian cancer, a PARPi response was linked to a low RAD51 score [70].

## 4. Outsourcing of HRD Analysis

FDA-approved diagnostic tests examine various components, resulting in different HR status definitions and, consequently, different treatment decisions [11] (Figure 3). As for the Myriad myChoice CDx and FoundationOne CDx tests, it is necessary to send the sample abroad for analysis, and 10-15% of samples provided inconclusive results.

The FDA has approved the Myriad myChoice CDx test as a companion diagnostic to select patients for therapy with olaparib and niraparib. Following the findings of the PRIMA and PAOLA1 trials, which employed it to identify HRD status in the first-line setting, this test is currently regarded as the standard [36]. The Myriad HRD score threshold was established by assessing HRD scores in a training cohort of 497 breast and 561 ovarian chemotherapy-naive cancers with known *BRCA1/2* status and determining a cutoff with 95% sensitivity to detect tumors with *BRCA1/2* mutations or *BRCA1* promoter methylation [71]. The GIS was bimodal in breast and ovarian tumor cohorts, with BRCA1/2 deficient tumors having a high GIS and *BRCA1/2* wild-type tumors having a low GIS. The threshold was set at 42 in this dichotomy to discriminate HRD tumors with 95% sensitivity. This number means that 5% of *BRCA1/2*-deficient tumors exhibit a GIS < 42, supporting the importance of parallel BRCA1/2 sequencing in the combined setting [67]. Notably, a fraction of tumors with unaltered *BRCA1/2* may also have elevated GIS, presumably due to the alteration of non-BRCA1/2 HRR genes [67]. Recently, the GIS threshold was changed from 42 to 33, demonstrating that several studies on alterations in HRR genes and their consequences on ovarian cancer are continuously evolving and being studied. The myChoice CDx tests from Myriad do not find HRR-dependent resistance mechanisms (like BRCA1/2, RAD51C/D, or PALB2 reversion mutations) or HRR-independent PARPi or BRCA1 epigenetic alterations.

Rucaparib uses the Foundation Medicine test FoundationOne CDx, which also includes analysis of an extensive panel of genes, encompassing *BRCA1/2* and many other HRR genes [60]. This test determines *BRCA1/2* mutation status for the companion diagnostic claim and assesses HR status, defined as the percentage of LOH regions within the tumor genome combined with the search for *BRCA1/2* alterations. The rate of genomic LOH is determined using next-generation sequencing technology. LOH-high is defined by a preset cutoff of 14% or higher [26]. 

## 5. HRD in a Laboratory Routine

There are significant differences in the method of HRD testing (causes versus effects; see Figure 2), but there are also variations in the assays that need to be examined. In addition to the companion diagnostics that have been approved by the FDA, other sequencing or single-nucleotide polymorphism (SNP)-based systems are being studied to measure genomic instability [11]. Several HRD assays are currently on the market, with the goal of offering HRD testing in diagnostic laboratories equipped with high-throughput NGS technology. Furthermore, European academic centres have made significant attempts to build a valid and feasible in-house HRD test to reproduce the Myriad MyChoice CDx results. A new study employed two academic genetic assays and a functional assay, the RAD51 foci, to determine HRD [72]. A total of 100 patients with high-grade OC who participated in the MITO16A/MaNGO-OV2 trial and were treated with carboplatin, paclitaxel, and bevacizumab as first-line therapy were studied [72]. The analysis revealed that multiple academic measures for HRD status are feasible, with good concordance with the current standard, the Myriad assay. Prospective validation is ongoing, and more functional and epigenetic evaluations, such as targeted methylation of HRR genes, will be available soon [72].

In addition, another academic network developed the “Leuven” HRD test as a part of the ENGOT European HRD initiative [73]. On a total of 468 ovarian cancer samples taken from the PAOLA-1/ENGOT-ov25 trial, researchers found a significant correlation between the results of the Leuven HRD and the Myriad myChoice PLUS test.

New assays evaluate HR by analyzing different alterations, and each proposes its cutoff (i.e., AmoyDX HRD Focus, Oncomine Comprehensive Assay Plus, SOPHiA DDM HRD Solution, and Illumina TruSight Oncology 500 HRD). Moreover, it is essential to compare these tests to the actual gold standard (BRCA1/2, GIS) in clinical trials to characterize better their predictive value and place it in the care pathway. The combination of several of these tests could provide a higher predictive value and needs to be deepened, bearing in mind that the set of results must be compatible with the starting times of the treatment [60]. 

To date, a plethora of in-house HRD testing assays is commercially available [74]. Regarding the implementation of this testing strategy in routine practice, this still represents an open challenge. In a previous experience, Fumagalli et al. evaluated the technical feasibility of the HRD Focus Assay, provided by AmoyDx (AmoyDx, Xiamen, China), that is able to detect *BRCA1/2* pathogenetic alterations and calculate HRD scores by adopting an optimized bioinformatic pipeline [75] on a retrospective series of n = 95 HGSCOC patients externally tested with Myriad’s myChoiceCDx solution [54]. Overall, a success rate of 84.2% for the HRD in-house testing strategy was assessed. In addition, a statistically significant concordance rate (97.3%) between these two methodological approaches was observed according to the *BRCA1/2* molecular assessment. Regarding HRD score calculation, a high concordance rate (87.5%) was also identified. Remarkably, the in-house testing approach highlighted an outstanding negative predictive value (100.0%) and an encouraging positive predictive value (83.3%) in comparison with an externalized solution. The failures were mostly caused by small biopsies or less-than-ideal DNA quality, caused by the fixative procedure and preanalytical conditions that affected DNA deamination and fragmentation [75]. Fumagalli et al. also compared the average reporting time of the in-house kit to Myriad, focusing on the accessibility and accuracy of in-house HRD testing. The median turn-around time for the kit in-house (AmoyDX HRD Focus) was 7 days from the testing order to the delivery of the report. This was much faster than the Myriad turn-around time of 18 days, which was also affected by processing and delivery logistics. The same test (AmoyDx HRD Focus) was used to analyze a cohort of 16 HGSC patients in a recent study by Magliacane and colleagues. The results showed excellent concordance with the Myriad assay and short turnaround times [76]. These aspects highlight the fact that HRD in-house testing approaches provide a technically comparable analytic solution for HRD calculation on real-world clinical samples [75].

One advantage of doing an in-house test is having control over the sample’s quality and amount, as well as the ability to pick a more appropriate sample if necessary. However, accounting for the technical heterogeneity that characterizes in-house HRD tests is critical. Variation in certain technical specifications (reference range, genomic figures analyzed, panel extension, for example) underscores the importance of harmonizing the analytical procedures before addressing internal HRD.

## 6. Current HRD Assay Limitations

There are some limitations to the existing HRD tests (Figure 4).

### 6.1. FFPE Material

The choice of tumor material for HRR gene screening is crucial. If faced with recurrence, it is preferable to use FFPE material, as the HRD phenotype of the tumor may change between the first diagnosis and recurrence. On the other hand, the quantity and quality of tissue stored in the FFPE block are sometimes inadequate and responsible for non-contributory specimens, and thus we should choose the material from the first diagnosis. This operation is not always possible, especially if patients are treated at different hospitals at diverse stages of the disease, so the only option, if the laboratory permits it, is to look for BRCA alterations at the germinal level. Furthermore, when analyzing FFPE material, the alterations discovered are frequently deamination artefacts or hyper-fragmentation of nucleic acids, which are not analyzed. Inappropriate tissue processing (delayed fixation and over-fixation) might affect sample quality and molecular test findings. 

It is suggested that molecular labs and pathology departments follow national or international standards, such as ISO 15189, to guarantee the quality standards during pre-analytical and analytical processes. 

### 6.2. Representative Tumour Area Selection

Representative tumor area selection and evaluation of the number of malignant cells, necrosis, and inflammatory components are critical for molecular tissue-based HRD assays [36]. In most cases, there must be at least 30% of the tumor for molecular methods to find a difference [61]. Due to a large number of inflammatory cell infiltrates in some HRD cancers, this can be hard to do.

### 6.3. Tumour Evolution Events

The clinical validity of HRD testing in ovarian cancer is now primarily assessed in terms of PARPi benefit rather than biological HRD status per se [36]. Beyond BRCA1/2 mutation, the principal open question is whether genomic scars are predictive biomarkers of responsiveness to platinum salts or PARPi [77]. The inability of genomic scar tests to capture tumor evolution events, such as the HRR activity restoration in response to therapy-selective pressure, is a current drawback [77]. Factors dynamically modulating the homologous recombination pathway and drug accumulation could hamper the predictive value of genomic scar HRD tests. In particular, there were no reports of secondary mutations or *BRCA1/2* reversions restoring homologous recombination [78]. A *BRCA* mutation may have originally imprinted a genomic HRD scar signature, but after reversal, the tumor would regain homologous recombination efficiency even if the HRD scar remained visible. It is especially critical in ovarian cancer, where approximately half of all platinum-resistant *BRCA* mutant tumors eventually recover *BRCA* function after platinum therapy [79,80]. Many BRCA1-independent mechanisms driving PARP inhibitor resistance are not identified in genomic scar HRD tests. Membrane transporters, for instance, are frequently involved in generating acquired or innate resistance [81]. As a result, tests that provide a functional assessment of HR in the tumor sample would be of clinical interest and advantage.

It would be ideal for precision medicine to set up an integrated strategy that combines clinical platinum sensitivity, genomic scar and mutational markers, and functional assays that show proof of HRD and the current functional ability of HRR throughout the clinical course.

### 6.4. Intratumor Heterogenicity

Successful solutions must address the issue of intratumor heterogeneity between the specimen biopsy site and additional metastatic locations, considering that there may be several cancer subclones present at any given time. Several studies examining the mutational landscape of different regions showed differences in the mutations observed depending on the location of the tumor sampled. These results suggest a spatial genomic heterogeneity within a single tumor, which could change the results of genomic biomarkers found in individual biopsies [82], such as the genomic scar.

The same tumor may be defined as “positive” or “negative” for HRD depending on the biopsy site because of “sampling error”, which is the sum of biological variation from biopsy to biopsy and the inherent technical noise of the method, including subtle differences in tissue composition between different biopsies (8). This genetic diversity can also be seen in various parts of the same tumor sample.

### 6.5. Alterations of Uncertain Significance (VUS)

The high rates of VUS in other HRR genes are probably due to the limited data available for interpretation concerning *BRCA1* and *BRCA2*. When it comes to analyzing HRR genes, different databases are referred to, and often conflicting or not entirely clear information is found. It happens because the effects of these alterations are not yet known. Some studies have found a high frequency of VUS in HRR genes in ovarian cancer patients, but they have also shown that two decades of testing and research on *BRCA1* and *BRCA2* have led to a significant decrease in VUS rates in *BRCA*, which are lower than in most other genes [83].

## 7. Conclusions

The analysis of HRD status is imperative in the therapeutic management of ovarian cancer patients. However, several preanalytical and analytical factors may impact its clinical testing in surgical pathology laboratories. 

In recent years, numerous HRD tests have been introduced on the market, but their clinical implementation is still far to be a routine practice. Multi-institutional efforts should determine the best approaches to guarantee adequate HRD testing for all patients with HGSOC.

## Figures and Tables

**Figure 1 jpm-13-00284-f001:**
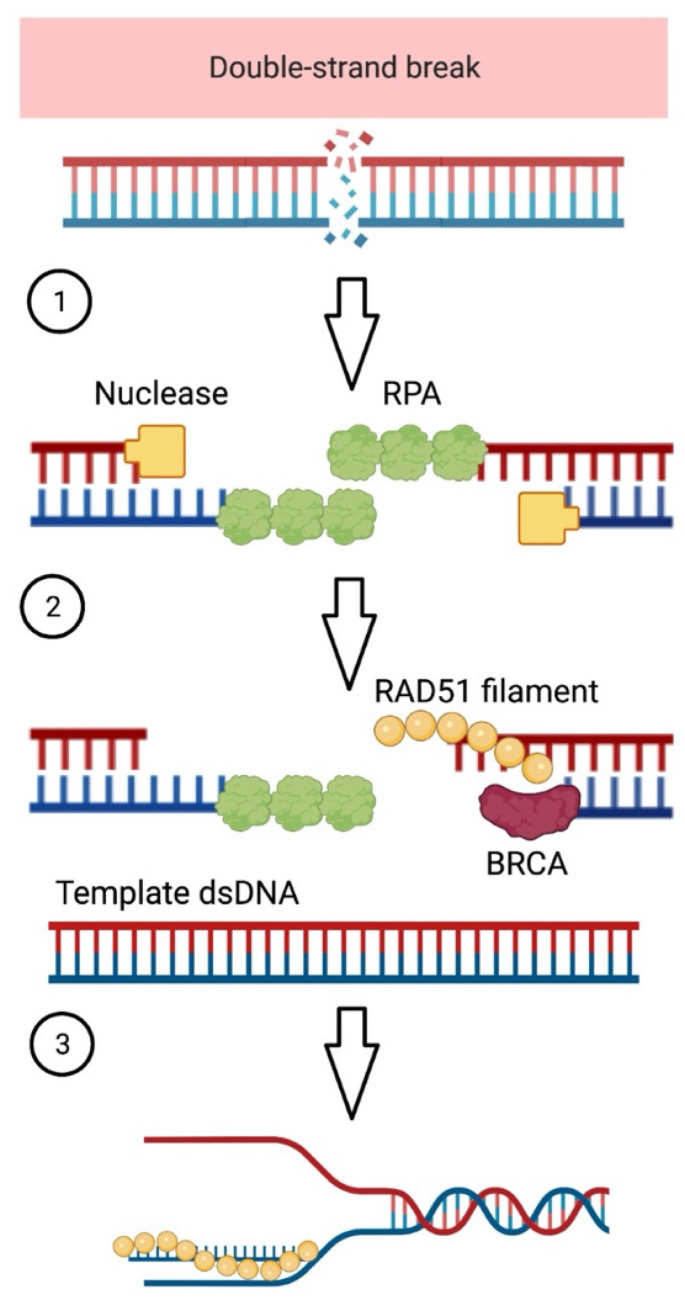
Breaks in double-stranded DNA (dsDNA) can be repaired through the homologous recombination repair (HRR) pathway. (**1**) The processing of double-stranded DNA (dsDNA) close to the broken ends by nuclease enzymes, which generates single-stranded DNA (ssDNA) overhangs, is the initial step. Replication protein A coats and protects the overhangs (RPA). (**2**) BRCA2 catalyzes the substitution of RPA with the RAD51 protein, resulting in the formation of a RAD51–ssDNA filament. (**3**) The filament invades a second DNA molecule with a matched sequence, which serves as a repair template. Created with BioRender.com.

**Figure 2 jpm-13-00284-f002:**
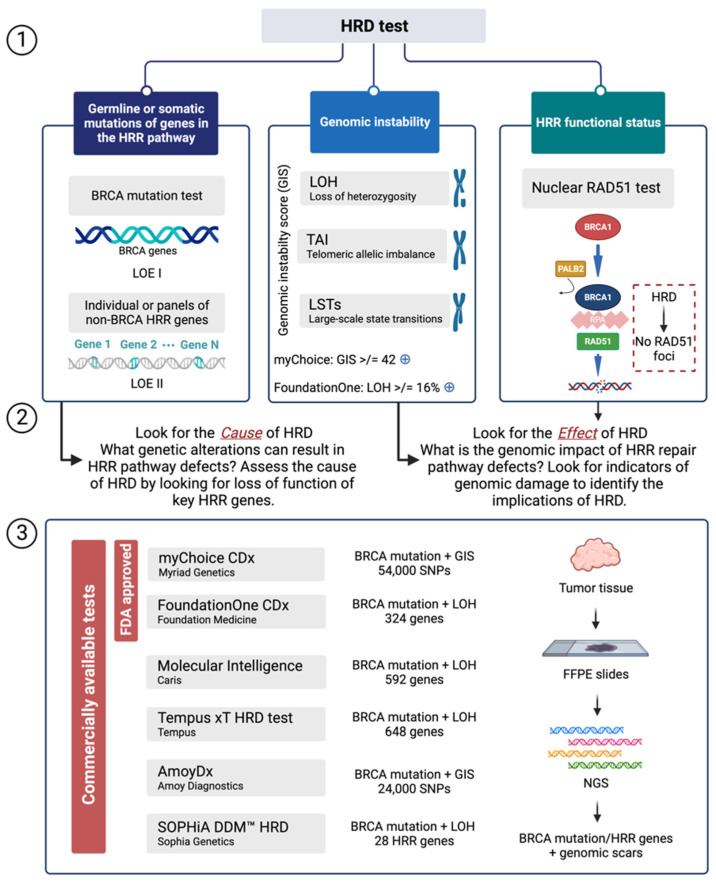
Approaches for testing HRD and commercially available assays. (**1**) HRD tests are classified into three types: germline or somatic mutations of genes in the HRR pathway, genomic instability, and HRR functional status testing. (**2**) There are two ways to detect HRD. The first is to look at the cause, specifically the loss of function of some essential HRR genes: BRCA mutation tests (LOE I) and individual or panels of non-BRCA HRR genes (LOE II). The alternative approach to identify HRD is to search for the effect of HRD, or the phenotype, and determine the consequences of HRD by looking at genomic damage. The Myriad myChoice CDx test uses the GIS score with a cutoff of 42, whereas the Foundation Medicine FoundationOne CDx test looks at LOH with a cutoff of 16. (**3**) Several laboratory tests have been developed as IVDs to detect HRD from NGS data. The analysis is performed on genomic DNA isolated from FFPE tumor tissue specimens. Two tests are FDA-approved IVDs: the Myriad myChoice CDx test and the Foundation Medicine FoundationOne CDx test. HRD: Homologous Recombination Deficiency; HRR: Homologous Recombination Repair; LOE: Level of Evidence; FFPE: Formalin-Fixed Paraffin-Embedded; NGS: Next Generation Sequencing; IVD: in vitro diagnostic device. Created with BioRender.com.

**Figure 3 jpm-13-00284-f003:**
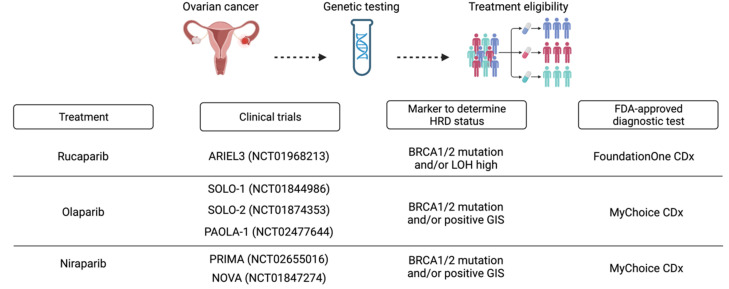
HRD analysis at diagnosis to determine treatment eligibility in ovarian cancer. Created with BioRender.com.

**Figure 4 jpm-13-00284-f004:**
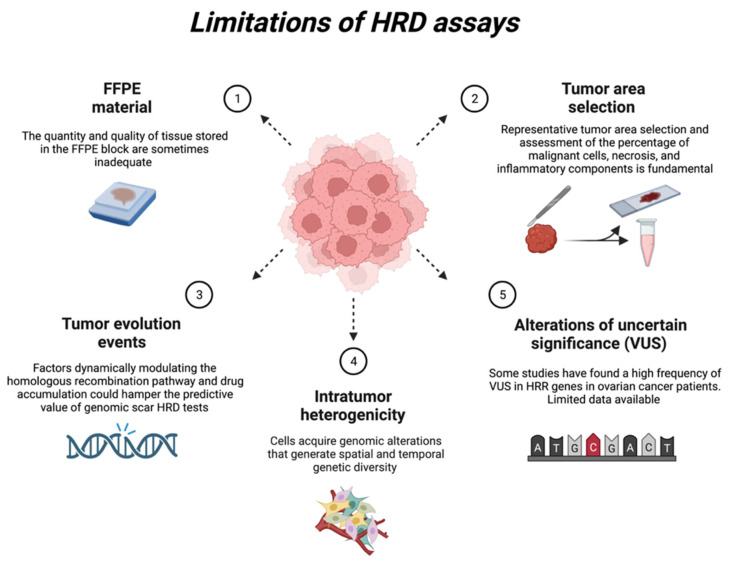
Current limitations of HRD tests. Created with BioRender.com.

**Table 1 jpm-13-00284-t001:** Most representative genes involved in the homologous recombination repair pathway [2,3,4,5,6].

ARID1A	EMSY	MSH2
ATM	FANCA	NBN
ATR	FANCC	PALB2
BRCA1/2	FANCE	PTEN
BARD1	FANCF	RAD50
BAP1	FANCD2	RAD51
BRIP1	FANCG	RAD51B
BLM	FANCI	RAD51C
CDK12	FANCL	RAD51D
CHEK1	H2AX	RAD54L
CHEK2	MRE11	TP53

**Table 2 jpm-13-00284-t002:** The three genomic scars were included in a genomic instability score.

Genomic scar	Description
Loss of heterozygosity (LOH)	One of the two alleles for a gene is lost, resulting in a homozygous cell. Failure of the remaining allele could result in the growth of malignant cells.
Telomeric allelic imbalance (TAI)	The proportion of alleles at the end of the chromosome (telomere) in a pair does not correspond, indicating that one chromosome has more alleles than the other.
Large-scale transitions (LSTs)	Breakpoints between regions of the chromosome cause discrepancies within the chromosome pair.

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
