# Peer review of "Homologous Recombination Deficiency in Ovarian Cancer: from the Biological Rationale to Current Diagnostic Approaches"

_jpm, 2023, doi:10.3390/jpm13020284_

Round 1

Reviewer 1 Report

In the article, the authors represent a review: Homologous recombination deficiency in ovarian cancer: from 2 the biological rationale to current diagnostic approaches.

Authors are encouraged to emphasize the current up-to-date diagnostic and therapeutic approaches/guidelines (NCCN, ESMO, and other national guidelines ..). It's important to mention when HRD is tested.

Authors are encouraged to do other changes in the article:

·        line 36 - mention all cancers gaining the significance of HRD

·        Table 1 – not all HRR genes are mentioned

·        RPA is a complex, not a gene

·        line 107 – please mentioned the original FDA citation, not the article combining all information

·        line 93 – also other articles emphasize this statement (example: https://pubmed.ncbi.nlm.nih.gov/35326583/)

·        not all studies/trials investigating HRD status are mentioned in the review, maybe the summary table of these studies will be better than the text

·        lines 200-205: maybe not relevant for this review? If yes, please cite the literature in line 205

·        line 212: Please mention that the HRD test is FDA-approved only for ovarian cancer (please check the literature)

·        Figure 2 – section 3: are all the tests available? Which test is IVD? For the mentioned test please recheck from which signs of genomic instability scores are calculated.

·        line 254 – there are data that offering tumor tissue testing first, followed by germline testing for PV/LPV detected in the tumor or additional NGS germline testing in cases of positive personal or family history for HBOC, would enable to detect nearly all PV/LPV as quickly as with parallel blood/tumor testing whilst greatly reducing the cost and workload involved (https://pubmed.ncbi.nlm.nih.gov/35326583/). The conclusion from the study: “Considering the facts that tumor testing enables detection of somatic and most germline PV/LPV, and that not all patients attend genetic counseling and testing, testing the tumor first seems to be a reasonable approach for detection of PV/LPV in HBOC genes in non-mucinous EOC patients. According to the study, there is a high probability that non-mucinous EOC patients without PV/LPV in these genes in tumors would be negative also after germline testing. For the purpose of accurate detection of germline and clinically actionable somatic PV/LPV in non-mucinous EOC patients, proposed workflow would be as follows: NGS genotyping of tumor samples first, followed in tumor positive cases by NGS or Sanger sequencing for determination of origin of PV/LPV (germline or somatic) and genetic counseling. In negative cases, the decision whether the patient is referred to genetic counseling and testing should be based on additional criteria such as personal and family history, histological type of tumor, and its molecular characteristics. Such an approach would enable a timely identification of nearly all germline and somatic PV/LPV, whilst reducing the workload and costs associated with simultaneous testing of blood and tumor samples.”

·        line 354 – please mentioned the appropriate literature

·        line 360 – 361 (in parentheses): some are tests and some are companies

·        Iine 376 – please mentioned the appropriate literature

I am looking forward to reading the review again.

Author Response

Response to Reviewer 1 Comments

In the article, the authors represent a review: Homologous recombination deficiency in ovarian cancer: from the biological rationale to current diagnostic approaches.

Point 1: Authors are encouraged to emphasize the current up-to-date diagnostic and therapeutic approaches/guidelines (NCCN, ESMO, and other national guidelines ..). It's important to mention when HRD is tested.

Response 1: We thank the reviewer for their careful reading of the manuscript and their constructive remarks. We have used the feedback to enhance and clarify the text. In the section "2. HRD in ovarian cancer," we have included a new paragraph to clarify when HRD is tested.

Point 2: Authors are encouraged to do other changes in the article:

  • line 36 - mention all cancers gaining the significance of HRD

Response 2: We have now mention all cancers gaining the significance of HRD.

Point 3:   Table 1 – not all HRR genes are mentioned.

Response 3: We have added some genes to the table and specified in the text that we report the most representative ones.

Point 4: RPA is a complex, not a gene.

Response 4: We apologize for the mistake and have now removed RPA from the table.

Point 5: line 107 – please mentioned the original FDA citation, not the article combining all information.

Response 5: We have now added the original FDA citation for each drug.

Point 6: line 93 – also other articles emphasize this statement (example: https://pubmed.ncbi.nlm.nih.gov/35326583/)

Response 6: We have now added a citation.

Point 7:  not all studies/trials investigating HRD status are mentioned in the review, maybe the summary table of these studies will be better than the text.

Response 7: We thank the reviewer for raising this critical issue. To respond to the reviewer's concern, we have added a new Table in the revised manuscript.

Point 8: lines 200-205: maybe not relevant for this review? If yes, please cite the literature in line 205.

Response 8: We appreciate the reviewer drawing our attention to this issue. We decided to mention the possible function of PARP inhibitors in immunotherapy, and we have now included a recent review on the subject.

Point 9: line 212: Please mention that the HRD test is FDA-approved only for ovarian cancer (please check the literature).

Response 9: We thank the reviewer for highlighting this point. We have now mentioned that the HRD test is FDA-approved only for ovarian cancer.

Point 10: Figure 2 – section 3: are all the tests available? Which test is IVD? For the mentioned test please recheck from which signs of genomic instability scores are calculated.

Response 10: Several laboratory tests have been developed as IVDs to detect HRD from NGS data. Two tests are FDA-approved IVDs: the Myriad myChoice CDx test and the Foundation Medicine FoundationOne CDx test. We have now added more clarity to this information in Figure 2, Section 3. We have also rechecked the type of genomic instability used to define the scores.

Point 11:   line 254 – there are data that offering tumor tissue testing first, followed by germline testing for PV/LPV detected in the tumor or additional NGS germline testing in cases of positive personal or family history for HBOC, would enable to detect nearly all PV/LPV as quickly as with parallel blood/tumor testing whilst greatly reducing the cost and workload involved (https://pubmed.ncbi.nlm.nih.gov/35326583/). The conclusion from the study: “Considering the facts that tumor testing enables detection of somatic and most germline PV/LPV, and that not all patients attend genetic counseling and testing, testing the tumor first seems to be a reasonable approach for detection of PV/LPV in HBOC genes in non-mucinous EOC patients. According to the study, there is a high probability that non-mucinous EOC patients without PV/LPV in these genes in tumors would be negative also after germline testing. For the purpose of accurate detection of germline and clinically actionable somatic PV/LPV in non-mucinous EOC patients, proposed workflow would be as follows: NGS genotyping of tumor samples first, followed in tumor positive cases by NGS or Sanger sequencing for determination of origin of PV/LPV (germline or somatic) and genetic counseling. In negative cases, the decision whether the patient is referred to genetic counseling and testing should be based on additional criteria such as personal and family history, histological type of tumor, and its molecular characteristics. Such an approach would enable a timely identification of nearly all germline and somatic PV/LPV, whilst reducing the workload and costs associated with simultaneous testing of blood and tumor samples.”

Response 11: We thank the reviewer for highlighting this point. We have now added this evidence to the text.

Point 12:      line 354 – please mentioned the appropriate literature.

Response 12: We have now added the appropriate literature.

Point 13: line 360 – 361 (in parentheses): some are tests and some are companies

Response 13: We apologize for the confusion. We have now removed the companies and left the test names.

Point 14:     Iine 376 – please mentioned the appropriate literature

Response 14: We have now added the appropriate literature.

Reviewer 2 Report

Mangogna and coworkers provide an extensive review of the technical aspects and challenges related to homologous recombination deficiency in ovarian cancer. The authors demonstrated efficiency in the search and compilation of a series of studies, but minor suggestions are present below to increase the quality of the paper:

A) There is an excessive description of the results of each study reviewed in the manuscript, with many numbers throughout the text, this makes the reading truncated. A new table compiling the PFS, HR, and CI results for each study can be included along with the critical contribution (so far) from each clinical trial, so the text is more fluid with fewer numbers and abbreviations to read.

B) Line 52: "Nonhomologous End Joining pathway." is written in capital letters.  Is an abbreviation missing?

C) Line 174: "placebo plus placebo maintenance" this is correct? 

D) After a series of clinical studies presented by the authors in section 2, the inclusion of a paragraph that summarizes the state of the art clarifies the idea about the topic for the readers. 

E) Line 207: "This Clinical tests for.." error of capital letter in the second word.

F) Line 227: only at this point in the manuscript do the authors provide the concept of "scars" already mentioned at the beginning of the text. This concept should be included after the first citation to avoid doubts about the term.

G) Line 340: "This There are significant..." revise.

H) Section 7: With many pharmaceutical tests to detect HDR,  the authors can cite in the text and conclusion the feedback challenge of these in-house tests. Is there monitoring of the results of these tests acquired and used by the population?

Author Response

Response to Reviewer 2 Comments

Mangogna and coworkers provide an extensive review of the technical aspects and challenges related to homologous recombination deficiency in ovarian cancer. The authors demonstrated efficiency in the search and compilation of a series of studies, but minor suggestions are present below to increase the quality of the paper:

Point 1: There is an excessive description of the results of each study reviewed in the manuscript, with many numbers throughout the text, this makes the reading truncated. A new table compiling the PFS, HR, and CI results for each study can be included along with the critical contribution (so far) from each clinical trial, so the text is more fluid with fewer numbers and abbreviations to read.

Response 1: We appreciate the reviewer's attentive study of the manuscript and helpful comments. As suggested by the reviewer, we have reorganized the section. We have now added a new Table 3 summarizing the clinical trials.

Point 2: Line 52: "Nonhomologous End Joining pathway." is written in capital letters.  Is an abbreviation missing?

Response 2: We apologize for the error; we have now added the abbreviation.

Point 3: Line 174: "placebo plus placebo maintenance" this is correct? 

Response 3: We apologize for the mistake: "…compared with chemotherapy plus placebo maintenance in patients with HGSOC.". In any case, this paragraph has been incorporated into a new Table 3. 

Point 4: After a series of clinical studies presented by the authors in section 2, the inclusion of a paragraph that summarizes the state of the art clarifies the idea about the topic for the readers.

Response 4: We have now added a paragraph and a reference summarizing the state of the art: “The SOLO-1 trial findings in 2018 led to olaparib being approved as a first-line maintenance treatment in patients with BRCA1/2 mutations by the EMA and the FDA, creating a new standard of care. Then, in 2019, the results of three phase III trials (PRIMA, PAOLA-1, and VELIA) that looked at the use of first-line PARP inhibitors beyond patients with BRCA1/2 mutations and as combination strategies were presented. This resulted in the recent approval of niraparib maintenance independent of biomarker status as well as olaparib in conjunction with bevacizumab in HRD-positive advanced ovarian cancer [41]. Following response to platinum-based chemotherapy, there is a substantial and clinically relevant advantage to adding PARP inhibitor maintenance treatment (alone or in com-bination with bevacizumab) in patients with HRD-positive tumors. Due to the low efficacy of PARP inhibitors in HR-proficient individuals and their worse prognosis, there is an urgent need for alternative therapeutic regimens in this patient subgroup [41].”

Point 5: Line 207: "This Clinical tests for.." error of capital letter in the second word.

Response 5: We apologize for the typing error; we have now corrected it.

Point 6: Line 227: only at this point in the manuscript do the authors provide the concept of "scars" already mentioned at the beginning of the text. This concept should be included after the first citation to avoid doubts about the term.

Response 6: We thank the reviewer for raising this critical issue. To respond to the reviewer’s concern, we have added the concept of “scars” after the first citation.

Point 7: Line 340: "This There are significant..." revise.

Response 7: We apologize for the typing mistake; we have now revised it.

Point 8: Section 7: With many pharmaceutical tests to detect HDR,  the authors can cite in the text and conclusion the feedback challenge of these in-house tests. Is there monitoring of the results of these tests acquired and used by the population?

Response 8: We thank the reviewer for raising this critical issue. We have now implemented the text in Section 5: “To date, a plethora of in-house HRD testing assays is commercially available [74]. As regards the implementation of this testing strategy in routine practice, this still represents an open challenge. In a previous experience, Fumagalli et al. evaluated the technical feasibility of the HRD Focus Assay provided by AmoyDx (AmoyDx, Xiamen, China), able to detect BRCA1/2 pathogenetic alterations and calculate HRD scores by adopting an optimized bioinformatic pipeline [75], on a retrospective series of n = 95 HGSCOC patients externally tested with Myriad's myChoiceCDx solution [54]. Overall, a success rate of 84.2% for the HRD in-house testing strategy was assessed. In addition, a statistically significant concordance rate (97.3%) between these two methodological approaches was observed according to the BRCA1/2 molecular assessment. Regarding HRD score cal-culation, a high concordance rate (87.5%) was also identified. Remarkably, the in-house testing approach highlighted an outstanding negative predictive value (100.0%) and an encouraging positive predictive value (83.3%) in comparison with an externalized solu-tion. The failures were mostly caused by small biopsies or less-than-ideal DNA quality, caused by the fixative procedure and preanalytical conditions that affected DNA de-amination and fragmentation [75]. Fumagalli et al. also compared the average reporting time of the in-house kit to Myriad, focusing on the accessibility and accuracy of in-house HRD testing. The median turn-around time for the kit in-house (AmoyDX HRD Focus) was 7 days from the testing order to the delivery of the report. This was much faster than the Myriad turn-around time of 18 days, which was also affected by processing and de-livery logistics. The same test (AmoyDx HRD Focus) was used to analyze a cohort of 16 HGSC patients in a recent study by Magliacane and colleagues. The results showed excellent concordance with the Myriad assay and short turnaround times [76]. These aspects highlight the fact that HRD in-house testing approaches provide a technically comparable analytic solution for HRD calculation on real-world clinical samples [75].”
